# The Financial Burden of Opioid-Related Abuse among Surgical and Non-Surgical Patients in Florida: A Longitudinal Study

**DOI:** 10.3390/ijerph18179127

**Published:** 2021-08-30

**Authors:** Jing Xu, Nazik M. A. Zakari, Hanadi Y. Hamadi, Sinyoung Park, Donald Rob Haley, Mei Zhao

**Affiliations:** 1Department of Health Administration, Brooks College of Health, University of North Florida, Jacksonville, FL 32224, USA; jasper.xu@unf.edu (J.X.); sinyoung.park@unf.edu (S.P.); rhaley@unf.edu (D.R.H.); mzhao@unf.edu (M.Z.); 2College of Applied Sciences, Al Maarefa University, Riyadh 11597, Saudi Arabia; nzakari@mcst.edu.sa

**Keywords:** opioids, opioid abuse, hospital, surgical, non-surgical, economic burden, United States

## Abstract

Florida is one of the eight states labeled as a high-burden opioid abuse state and is an epicenter for opioid use and misuse. The aim of our study was to measure multi-year total room charges and costs billed for opioid abuse-related events and to compare the costs of inpatient opioid abusers and non-opioid abusers for Florida hospitals from 2011 to 2017. We constructed a retrospective case-control longitudinal study design on inpatient administrative discharge data across 173 hospitals. Opioid abuse was defined using both ICD-9-CM and ICD-10-CM systems. We found a statistically significant association between opioid abuse diagnosis and total room charge. On average, opioid abuse status increased the room charges by 8.1%. We also noticed year-to-year variations in opioid abuse had a remarkable influence on hospital finances. We showed that since 2015, the differences significantly increased from 4–5% to 13–14% for both room charges and cost, which indicates the financial burden due to opioid abuse becoming more frequent. These findings are important to policymakers and hospital administrators because they provide crucial insight into Florida’s opioid crisis and its economic burden on hospitals.

## 1. Introduction

Opioids are defined by the National Institute on Drug Abuse as a class of drugs that include the illegal drug heroin, synthetic opioids such as fentanyl, and pain relievers [1]. Opioid abuse has risen to epidemic levels in the United States and was declared a public health emergency in 2017 [2,3]. Around 2 million people aged 12 or older suffered an opioid use disorder in 2018 [1], and 47,600 deaths were attributed to opioid-related abuse in 2017, highlighting increases in deaths among non-Hispanic blacks and Hispanics [4]. Fortunately, opioid-involved death rates decreased by 2.0% from 2017 to 2018, which may be explained by efforts to reduce high-dose opioid prescribing practices [4]. Since the mid-2000s, Florida has been called an epicenter of the opioid epidemic, with opioid-related deaths increasing by 80% from 2003 to 2009 [5]. For this reason, the Prescription Drug Monitoring Program (PDMP) was initiated in September 2011. Although PDMP implementation helped opioid-related deaths decrease overall [6], the overdose death rate in Florida remained high at 15.8 per 100,000 in 2018, with the national average being 14.6 per 100,000 [7].

Opioid-related abuse has caused a marked increase in economic burdens, such as increased healthcare costs, workplace costs including reduced wages, lower employment, loss of worker productivity, and criminal justice costs [8,9]. The total economic burden was estimated to be $1021 billion at the national level and $68,444 million in Florida alone in 2017 [10,11]. The healthcare costs of opioid abuse in particular, calculated based on payer reimbursements to providers, have risen over time, mainly driven by the treatment of substance abuse, rehabilitation, inpatient, and emergency department (ED) costs [12,13].

The number of hospital charges and ED visits doubled from 2014 to 2017 due to opioid abuse, dependence, or overdoses [14]. This increasing trend in opioid abuse and misuse may be due to a more acute care population of patients in hospitals resulting in an increased prescribing of opioids [15]. Opioid abusers had higher healthcare utilization compared to the average population, including inpatient, outpatient, ED visits, and rehabilitation facility utilization [13,16,17]. Prior studies utilized medical and pharmacy claims data from 1998 to 2002 of self-insured employer health plans and found that opioid abusers are 12 times, four times, and 63 times more likely to have an inpatient stay, an ED visit, or an outpatient visit, respectively, compared with non-abusers matched by age, gender, employment status, and census region [17]. Opioid abusers are also more likely to have higher 30-day readmission rates and longer in a drug rehabilitation facility longer relative to matched non-abusers [13,18]. Similarly, prescription drug utilization was higher for opioid abusers than controls [19]. These phenomena could be explained by opioid abusers having higher rates of baseline comorbidities, such as pain and mental disorders, and higher chances of having other substance abuse disorders than non-abusers.

Higher healthcare utilization has increased healthcare costs for opioid abusers directly [20]. Numerous prior studies have examined healthcare costs associated with opioid abuse. Healthcare cost estimates vary across studies depending on study population characteristics, such as being insured, being on Medicare or Medicaid, specific healthcare events such as overdoses, and methods for measuring costs. However, those studies commonly indicate that healthcare costs tend to be higher for opioid abusers compared to non-abusers [12,13,16,17,21,22,23].

As such, for a population that is privately insured, annual healthcare costs were $18,000 higher for opioid abusers patients than non-abusers [17]. A separate study analyzing administrative claims from self-insured recipients found that the average annual healthcare cost of opioid abusers per patient was $20,343, compared to $9716 for controls [13]. A second analysis was conducted using a different administrative claims dataset with the same matching approach and had similar results [16]. A recent study also confirmed that opioid abusers had $14,810 higher healthcare costs annually [12]. A national study that analyzed Medicaid-covered opioid abusers reported that opioid abusers accumulated approximately triple the healthcare costs of non-abusers [21]. Some studies analyzed hospital charges related to opioid abuse. Higher hospital charges were driven by an increase in the utilization of ED services and inpatient admissions for opioid abusers, which are especially high among Medicare or Medicaid-covered opioid abusers [24,25].

Much of the previous literature analyzed healthcare utilization and direct costs associated with opioid abuse, using commercially available insurance claims data, managed care plan claims data, and Medicare/Medicaid claims data. However, there is limited literature investigating the financial burden of hospital services for opioid abusers from a hospital perspective. Therefore, the purpose of this study was to examine the difference in financial cost, measured by total room charge and cost, between opioid and non-opioid abusers across all Florida inpatient hospital discharges from 2011 to 2017.

### Conceptual Framework

Based on the literature and the factors that impact opioid use hospital treatment expenditure, we developed the following framework guided by the framework developed in 2019 by Leslie, et al. [26]. The framework focuses on the path of patients with pain conditions. These patients may begin to use prescribed opioid medication and may become addicted which can result in an opioid-abuse diagnosis. Treatment is typically sought out and initiated in the ED and leads to further healthcare services including but not limited to inpatient services (e.g., hospitalizations and residential rehabilitation services). Our framework describes the financial path that opioid abuse hospital treatment and how it may drive up hospital expenditure. To evaluate the difference in financial cost, measured by total room charge and cost, between opioid and non-opioid abusers, we first followed patients with opioid use disorder treatment and their entry point to the hospital system. This entry point is often initiated in the emergency department and leads to further healthcare service use. In this model, we used hospital total room charge and cost from administrative claims data. We believe that patients with opioid use disorder may require additional treatment for secondary conditions presented due to the opioid disorder. We only include hospital expenditures and do not include insurance reimbursement payments or patient out-of-pocket payments.

## 2. Methods

### 2.1. Data Sources

The Florida Inpatient Discharge (FID) dataset was utilized in this study to examine the financial impact associated with opioid use. Inpatient discharges for all Florida hospitals from 2011 to 2017 were obtained from the Agency for Health Care Administration’s Center for Health Information and Transparency. The data consisted of over 13 million de-identified medical claims records for each inpatient encounter and recorded patients’ medical, financial, and demographic information. The data recoded up to 30 diagnosis and procedural codes per encounter, making it a comprehensive dataset. The data also provided a hospital identifier known as the Medicare Provider Number. Using this identifier, the dataset was linked to the 2011–2017 American Hospital Association (AHA) annual survey to populate hospital-related information. The AHA collects information on all U.S. hospitals’ organizational structure, financing, workforce, and care delivery.

### 2.2. Variables and Measures

#### 2.2.1. Outcome Variable

Our primary outcome variable was a single inpatient hospital stay-related charge for all encounters between 2011 and 2017.

#### 2.2.2. Case and Control Groups

In this study, each encounter was assigned to be in either a case (opioid abuse) or control (non-opioid abuse) group based on whether it had a diagnosis of opioid abuse. Our inclusion criteria for both control and treatment groups were patients admitted to a hospital for treatment between 2011 and 2017. We included in the control group patients without an opioid diagnosis. In the treatment group we included patients with at least 1 opioid abuse diagnosis. We excluded any patients with missing data. Opioid abuse diagnosis was determined using both primary or secondary ICD-9-CM and ICD-10-CM codes presented in Table 1. The codes were identified and extracted based on prior research that used ICD-9-CM and ICD-10-CM codes and the Healthcare Cost and Utilization Project (HCUP) [27,28].

### 2.3. Matching Procedure

Due to the low incidence rate of the opioid abuse case in the overall sample (<2%) and a high absolute number of opioid abuse cases across the study period, [29] a propensity score method (PSM) was applied to create homogeneous matched samples for the final analysis [30]. The PSMATCH procedure from SAS 9.4 (Statistical Analysis System, Cary, North Carolina, United States) was applied to complete the propensity score matching.

For each opioid abuse case, the matching chose the non-abuse case that minimizes the difference between the logits of the propensity scores. The “greedy nearest neighbor” matching algorithm was used with a caliper width of 0.25. The matching procedure was independently applied by yearly quarters (28 quarters). All opioid encounters were matched with non-abuse encounters on 1 to 1 ratio by patient age, gender (male or female), race (White, African American/Black, Latino/Hispanic, Asian, or other), primary surgical procedure (yes or no), length of stay, and Elixhauser comorbidity index, among which patient gender was required match exactly. Surgical procedures were defined using Healthcare Cost and Utilization Project (HCUP) Surgery Flags software for ICD-9-CM and ICD-10-CM [27,28]. Furthermore, we used a two-digit surgical procedure code prefixes which are listed in Table 2. We also matched specific hospital characteristics that have been previously identified as having an important impact on hospital charges across the entire study period. These hospital characteristics were hospital location (rural or urban), size (small < 100, medium < 400, and large 400+ inpatient beds), ownership (for-profit, not-for-profit, or government), and teaching affiliation (major, minor, or non-teaching).

The room charge and estimated cost (room charge multiplied by the cost-to-charge ratio) were analyzed separately by generalized linear models [31]. with lognormal distribution and identity link function using SAS 9.4 [32]. The experiment factor—opioid abuse status—was included in the model as a fixed covariate [33]. All the matching variables mentioned in the previous section were also included as covariates. Besides the main effect of opioid abuse on both financial metrics, we also examined the impacts by year and surgical status.

## 3. Results

The distribution of room charges and hospital cost are provided in Table 3 by year. The distribution shows that for room charges the largest difference between our control and the treatment group was in 2015, while the smallest difference was in 2014. For hospital costs the largest difference was reported in 2015, with a 20.58% difference between treatment and control groups, while the smallest difference was in 2012. The final sample sizes and matching statistics (total absolute differences) are provided in Table 4. The analysis of room charges and cost per encounter was provided in Table 5. It shows, on average, that opioid abuse status increases room charges by 8.1% with a 95% confidence interval (7.1%, 9.1%), and increases the room cost by 8.6% with a 95% confidence interval (7.6%, 9.6%) per encounter.

We also examined how the differences varied year to year, as well as whether the differences varied between surgical and non-surgical patients. Therefore, we conducted a longitudinal analysis with stratified comparisons. It was also noticed that the year-to-year variations of the influence of opioid abuse on hospital finances are remarkable. Compared with the control groups, the increases in in-room charges and cost are consistently higher from 2011 to 2017. The differences are about 5–8% for the room charge and 4–9% for the cost from 2011 to 2014. From 2015 onward, the differences significantly increased to 13–14% for both room charges and cost, which indicates the financial burden due to opioid abuse is increasing with time.

The surgical and non-surgical subgroups are further examined, and the same pattern over the past seven years was found. The variation in the non-surgical groups is significantly smaller, which makes the differences statistically significant every year. On the other hand, due to the nature of various surgical procedures, the variation in the surgical group is higher, especially from 2011 to 2014. Regardless, the increases in in-room charges and cost in both subgroups have been significantly higher since 2015, indicating that this is a systematic issue that impacts all inpatient populations in Florida.

## 4. Discussion

The findings of this study indicate that the abuse of opioids for Florida patients could lead to a significant financial burden, with the abusing patients incurring 8% to 10% more in both charges and costs compared with their non-abusing counterparts. This is consistent with the literature that opioid overdoses cost U.S. hospitals an estimated $11 billion annually [34]. The Centers for Disease Control and Prevention estimate that the total “economic burden” of prescription opioid misuse alone in the United States is $78.5 billion a year, including the costs of healthcare, lost productivity, addiction treatment, and criminal justice involvement [35]. As the front lines of the opioid epidemic, hospitals are confronted with increasing pressures on capacity and resources related to the diagnosis and treatments of opioids [36]. Opioid-related hospital use, including emergency department visits and inpatient stays, increased significantly over the past decade, which contributes significantly to hospital costs. Opioid abuse and misuse in hospitals may be due to more acute patients in hospitals and increased use and of opioids as a result [15]. The reasons why US hospital opioid use increased include the use of opioids in the management of chronic pain; CMS’s reimbursement policy, which is tied to patient care experience measures; and the 2000 Joint Commission’s report, as potential drivers for inappropriate opioid administration in hospitals [37]. Studies suggest that clinicians may still rely on outdated Joint Commission standards that defined pain as a “fifth vital sign,” leading to overaggressive pain management during hospitalization [38]. In addition, clinicians may inappropriately order opioids due to pressures to obtain better patient care experience scores. CMS decided to temporarily remove two pain-care patient care experience questions from the hospital reimbursement formula starting in 2018 [39]. The CMS plans to add questions on communication about pain care instead. Many hospitals have already responded to the opioid epidemic by changing their prescription of opioid medication practices and the settings in which they are prescribed to reduce these societal and internal costs. There is an increased need for hospitals and clinicians to develop evidence-based prescribing guidelines, encourage safe opioid disposal, and develop patient education materials. In addition, since the prevalence of opioid abuse is not only the responsibility of the hospitals, other community stakeholders should also be involved to reduce the stigma associated with this epidemic.

For example, it is difficult to decrease the cost of healthcare without identifying policies to reduce the incidence of those patients suffering from opioid abuse. As a result, policies that limit the prescribing of opioids or change practices of care by educating providers on the negative impacts of opioid abuse on patient outcomes and costs should be implemented and strengthened. The effect of health insurance on opioid abuse and misuse has been under debate. Some believe that access to health insurance increased access to prescribed opioids. However, policies are increasingly being implemented for insurance companies to monitor opioid usage more closely and to educate providers on proper opioid prescription policies [40]. For example, under the ACA the expansion of health insurance was associated with meaningful reductions in opioid-related hospital use, and the proactive utilization of management care for opioid use disorder among the Medicaid expansion segment [41]. While the effect of health insurance has increased access to prescribed opioids, providers and insurance companies can reduce the need for opioids through best practices to improve health, closer monitoring of opioid usage, proactive prescribing of mental health services, and drug treatment [40]. Furthermore, at-risk patients should be educated on opioid addiction and its potential burden in terms of their health and costs. In order to lower healthcare costs, policies should be considered to improve patient, family, and caregiver engagement and encourage them to question the prescribing of opioid medications. Policies that encourage pharmacists to question providers for potentially inappropriate opioid prescription quantities should be encouraged and strengthened [42].

Finally, the findings presented in our study are especially important to providers and hospital administrators who provide uncompensated care to patients who suffer from opioid abuse. Administrators should consider assessing their provider opioid education programs and opioid prescribing trends. In addition, providers should assess their patient opioid education programs on patient outcomes to decrease costs and uncompensated care. Therefore, it is important for hospital administrators to consistently improve and proactively identify solutions to decrease opioid abuse and measurably decrease the cost of care.

## 5. Conclusions

This study provides important insights into Florida’s opioid crisis and its financial impact on patients and providers. Our results indicate that there is a significant association between opioid abuse and an increase in both charges and costs in Florida hospitals. Policymakers should understand the impact of this association when developing policies to decrease healthcare costs. Furthermore, insurance companies and pharmacy benefit managers are in a unique position to monitor opioid usage for the insured since they manage prescriptions claims. Effective policies should be implemented by health plans that identify appropriate care and prevention strategies and determine the most effective methods for realizing health outcomes to improve population health and optimize reimbursement, including policies such as value-based purchasing.

## Figures and Tables

**Table 1 ijerph-18-09127-t001:** ICD-9-CM And ICD-10-CM Opioid-Abuse Diagnosis Codes.

ICD-9-CM Code	ICD-10-CM Code	Description of Opioid Abuse
305.50		Unspecified
305.51		Continuous
305.52		Episodic
	F11.10	Uncomplicated
	F11.120	With intoxication, uncomplicated
	F11.121	With intoxication delirium
	F11.122	With intoxication with perceptual disturbance
	F11.129	With intoxication, unspecified
	F11.14	With an opioid-induced mood disorder
	F11.150	With opioid-induced psychotic disorder with delusions
	F11.151	With opioid-induced psychotic disorder with hallucinations
	F11.159	With opioid-induced psychotic disorder, unspecified
	F11.181	With opioid-induced sexual dysfunction
	F11.182	With opioid-induced sleep disorder
	F11.188	With other opioid-induced disorder
	F11.19	With unspecified opioid-induced disorder

**Table 2 ijerph-18-09127-t002:** Listing of HCUP Prefix Two-Digit Surgical Procedures.

Principal Procedure	Description of Surgical Operation
ICD-9-CM	
01–05	Nervous System
06–07	Endocrine System
08–16	Eye
18–20	Ear
21–29	Nose, Mouth, And Pharynx
30–34	Respiratory System
35–39	Cardiovascular System
40–41	Hemic And Lymphatic System
42–54	Digestive System
55–59	Urinary System
60–64	Male Genital Organs
65–71	Female Genital Organs
74	Cesarean Section and Removal of Fetus
76–84	Musculoskeletal System
85–86	Integumentary System
ICD-10-CM	
00	Central Nervous System and Cranial Nerves
01	Peripheral Nervous System
02	Heart and Great Vessels
03	Upper Arteries
04	Lower Arteries
05	Upper Veins
06	Lower Veins
07	Lymphatic and Hemic Systems
08	Eye
09	Ear, Nose, Sinus
0B	Respiratory System
0C	Mouth and Throat
0D	Gastrointestinal System
0F	Hepatobiliary System and Pancreas
0G	Endocrine System
0H	Skin and Breast
0J	Subcutaneous Tissue and Fascia
0K	Muscles
0L	Tendons
0M	Bursae and Ligaments
0N	Head and Facial Bones
0P	Upper Bones
0Q	Lower Bones
0R	Upper Joints
0S	Lower Joints
0T	Urinary System
0U	Female Reproductive System
0V	Male Reproductive System
0W	Anatomical Regions, General
0X	Anatomical Regions, Upper Extremities
0Y	Anatomical Regions, Lower Extremities

**Table 3 ijerph-18-09127-t003:** Descriptive Statistics of Treatment (Opioid Abuse Status) and Control Groups’ (Non-Opioid Abuse Status) Room Charges and Hospital Costs.

Room Charges				
Year	Treatment Group (Abuse = 1)	Control Group (Abuse = 0)	Difference (Treatment—Control)	Percent Change
2011	$1211.87	$10,513.54	$1604.33	15.26%
2012	$19,251.52	$17,061.79	$2189.73	12.83%
2013	$13,967.19	$11,956.54	$2010.65	16.82%
2014	$7787.55	$6912.42	$875.13	12.66%
2015	$16,150.42	$13,732.43	$2417.98	17.61%
2016	$8768.44	$7719.53	$1048.91	13.59%
2017	$9628.66	$8343.53	$1285.13	15.40%
Hospital Cost				
Year	Treatment Group (Abuse = 1)	Control Group (Abuse = 0)	Difference (Treatment—Control)	Percent Change
2011	$2484.52	$2153.24	$331.29	15.39%
2012	$3568.87	$3178.65	$390.22	12.28%
2013	$2462.00	$2134.73	$327.27	15.33%
2014	$1318.17	$1173.28	$144.89	12.35%
2015	$2690.27	$2231.16	$459.11	20.58%
2016	$1505.16	$1320.95	$184.21	13.95%
2017	$1456.34	$1253.02	$203.32	16.23%

**Table 4 ijerph-18-09127-t004:** Descriptive Statistics of Treatment (Opioid Abuse Status) and Control Groups (Non-Opioid Abuse Status) Sampling.

Case-Matching Sample Size			
Year	Treatment Group (Abuse = 1)	Control Group (Abuse = 0)	Total Absolute Difference
2011	5398	5398	0.086
2012	5400	5400	0.091
2013	6343	6343	0.133
2014	7200	7200	0.089
2015	10,119	10,119	0.156
2016	8033	8033	0.131
2017	12,158	12,158	0.216

**Table 5 ijerph-18-09127-t005:** Data Analysis of Percentage Differences between Treatment (Opioid Abuse) and Control Groups (Non-Opioid Abuse) Associated Room and Hospital Costs.

Percent Difference and 95% CI in Room Charges			
Year	Overall	Surgical Group	Non-Surgical Group
Overall	8.1 (7.1, 9.1) **		
2011	5.7 (−6.1, 19.0)	7.4 (4.4, 10.6) **	3.9 (−17.9, 31.6)
2012	8.1 (−5.3, 23.3)	4.8 (1.6, 7.5) **	11.7 (−14.1, 45.3)
2013	4.9 (−6.4, 17.5)	4.6 (1.9, 7.4) **	5.1 (−16.1, 31.7)
2014	7.1 (−3.9, 19.2)	6.8 (4.2, 9.4) **	7.4 (−13.3, 32.9)
2015	12.9 (8.9, 17.0) **	11.0 (8.6, 13.4) **	14.8 (7.1, 23.0) **
2016	13.0 (10.0, 16.2) **	11.1 (8.2, 14.1) **	15.0 (9.7, 20.1) **
2017	8.4 (6.1, 10.8) **	6.6 (4.3, 8.9) **	10.3 (6.2, 14.5) **
Percent Difference and 95% CI in Hospital Cost (Room Charges Multiplied by the Cost-to-Charge Ratio)			
Overall	8.6 (7.6, 9.6) **		
2011	4.4 (−7.6, 17.9)	8.8 (5.7, 12.0) **	0.1 (−21.4, 27.5)
2012	9.4 (−4.5, 25.3)	3.8 (0.8, 6.8) **	15.3 (−12.0, 51.1)
2013	3.7 (−7.7, 16.6)	3.9 (1.2, 6.8) **	3.5 (−17.9, 30.4)
2014	9.2 (−2.3, 21.9)	6.2 (3.6, 8.9) **	12.2 (−9.9, 39.8)
2015	14.6 (10.4, 18.9) **	11.2 (8.8, 13.7) **	18.0 (9.9, 26.6) **
2016	13.3 (10.2, 16.6) **	12.2 (9.2, 15.3) **	14.4 (8.9, 20.2) **
2017	9.8 (7.4, 12.3) **	8.4(6.0, 10.8) **	11.3 (7.1, 15.7) **

Note: CI: Confidence interval; ** Significant at 0.05 level.

## Data Availability

Restrictions apply to the availability of these data. Data was obtained from the Agency For Health Care Administration in Florida and are available at https://www.floridahealthfinder.gov/Researchers/OrderData/order-data.aspx (accessed on 16 July 2021) with the permission of the Agency For Health Care Administration.

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
