# Peer review of "The Financial Burden of Opioid-Related Abuse among Surgical and Non-Surgical Patients in Florida: A Longitudinal Study"

_ijerph, 2021, doi:10.3390/ijerph18179127_

Round 1

Reviewer 1 Report

Review of manuscript entitled: “The Economic Burden of Opioid-Related Abuse Among Surgical and Non-Surgical Patients: A Longitudinal Study” authored by Jing Xu, Nazik M.A. Zakari, Hanadi Hamadi, Sinyoung Park, D. Rob Haley, Mei Zhao

In my opinion, despite important topic, current form of the presented manuscript cannot be published as an “Article”. I would recommend to extend the analysis (since it has potential) or change the article type to “Short communication”.

Major concerns:

  • Title – please consider adding the investigated geographical region since in its present form title suggests that its worldwide study
  • Introduction
    • Subsections are not necessary in introduction, please try to smoothly merge them
    • In my opinion hypothesis presented by authors that patients with opioid abuse require higher financial burden than patients without opioid abuse is pretty obvious – hence from reading the introduction I do not see clear point of this study
  • Methods/Results
    • I am quite confused about “non-opioid abusers”, does it mean that patients chosen to this group could be alcohol abusers or were they free of any kind of abuse? If they were not abusing any substance I would recommend to change term “non-opioid abusers” to something else.
  • Results
    • I do not understand why presented results are so limited comparing to potential of the dataset. From methods section I read that authors included several variables regarding the patients (e.g. sex, race) and hospitals (e.g. location, size), hence results are only presented for non-abusers vs abusers. Including more detailed analysis of this dataset would increase the impact of the manuscript (for example, if in rural hospitals the difference in room charges between opioid abusers and non-abusers was higher/lower that in hospitals located in cities). In my opinion there is a minimum of 4-5 additional comparisons which are possible to compute and which will increase the outcome of your research. Or am I missing something?
  • Discussion
    • Very limited and short due to limited analysis of the dataset
  • I believe that “Policy implication” paragraph should be merged with discussion

Minor concerns:

  • Affiliations – position, ORCiD, fax number, telephone number are not necessary + unify font style/size
  • Abstract – please adjust to editor requirements (no headings) + word count
  • Line 47 – typo “.,”
  • Line 56 – please provide full description of ED abbreviation
  • Author contribution – repeated heading and divided into two parts

Author Response

Reviewer 1

In my opinion, despite important topic, current form of the presented manuscript cannot be published as an “Article”. I would recommend to extend the analysis (since it has potential) or change the article type to “Short communication”.

Thank you for your feedback, The analysis for this study is extensive in nature and the propensity matching technique used. We have added additional information on all the steps done on the analysis and data manipulation to justify the article type chosen.

Major concerns:

  • Title – please consider adding the investigated geographical region since in its present form title suggests that its worldwide study

Thank you, we have updated the study title to “The Financial Burden of Opioid-Related Abuse Among Surgical and Non-Surgical Patients in Florida: A Longitudinal Study”

  • Introduction
    • Subsections are not necessary in introduction, please try to smoothly merge them

Thank you, we have removed the subsections and have improved the paragraph transition.

    • In my opinion hypothesis presented by authors that patients with opioid abuse require higher financial burden than patients without opioid abuse is pretty obvious – hence from reading the introduction I do not see clear point of this study

Thank you, we have removed the hypothesis and provided a clear purpose statement.

  • Methods/Results
    • I am quite confused about “non-opioid abusers”, does it mean that patients chosen to this group could be alcohol abusers or were they free of any kind of abuse? If they were not abusing any substance, I would recommend to change term “non-opioid abusers” to something else.

Thank you for your feedback, the non-opioid abusers were free only from opioid abuse. Opioids drugs that are prescribed to treat pain and are not substance such as alcohol. Our terminology adheres to the current body of literature base on these citations below. We have added clarifying statement regarding the definition of non-opioid abusers in the introduction and stated that “Opioids are defined by the National Institute on Drug Abuse as a class of drugs that include the illegal drug heroin, synthetic opioids such as fentanyl, and pain relievers.”

  1. Franke, P., Neef, D., Weiffenbach, O., Gänsicke, M., Hautzinger, M., & Maier, W. (2003). Psychiatric comorbidity in risk groups of opioid addiction: a comparison between opioid dependent and non-opioid dependent prisoners (in jail due to the German narcotics law). Fortschritte der Neurologie-psychiatrie, 71(1), 37-44.
  2. Jalali, M. R., Zargar, M., Salavati, M., & Kakavand, A. R. (2011). Comparison of early maladaptive schemas and parenting origins in patients with opioid abuse and non-abusers. Iranian journal of psychiatry, 6(2), 54.
  3. Vakharia, R. M., Donnally III, C. J., Rush III, A. J., Vakharia, A. M., Berglund, D. D., Shah, N. V., & Wang, M. Y. (2018). Comparison of implant survivability in primary 1-to 2-level lumbar fusion amongst opioid abusers and non-opioid abusers. Journal of Spine Surgery, 4(3), 568.

  • Results
    • I do not understand why presented results are so limited comparing to potential of the dataset. From methods section I read that the authors included several variables regarding the patients (e.g. sex, race) and hospitals (e.g. location, size), hence results are only presented for non-abusers vs abusers. Including more detailed analysis of this dataset would increase the impact of the manuscript (for example, if in rural hospitals the difference in room charges between opioid abusers and non-abusers was higher/lower that in hospitals located in cities). In my opinion there is a minimum of 4-5 additional comparisons which are possible to compute and which will increase the outcome of your research. Or am I missing something?

Thank you for your feedback, The analysis for this study is extensive in nature and we completed a propensity matching technique used is appropriate. We have added additional information on all the steps done in the analysis and data manipulating.

All the variables we have identified were used in the matching strategy step, so when we compare the differences, our sample in the treatment group is similar to that of the control group regarding the matching factors.

I do agree with you that there is a potential to make more stratified comparisons. In this paper, we chose to focus on the surgical status as a stratification factor. In addition, we checked the longitudinal effect by taking time into consideration.

We have added additional language to clarify the methodology in depth.

  • Discussion
    • Very limited and short due to limited analysis of the dataset
  • I believe that “Policy implication” paragraph should be merged with discussion

Thank you, we have measured the policy implication paragraph to the discussion.

Minor concerns:

  • Affiliations – position, ORCiD, fax number, telephone number are not necessary + unify font style/size

Thank you, we have removed the additional affiliation information and ensured font size uniformity.

  • Abstract – please adjust to editor requirements (no headings) + word count

Thank you. We have updated the abstract to adhere to the no-heading and 200-word count requirement. The abstract now reads, “The aim of our study was to measure multi-year total room charges and costs billed for opioid abuse-related events and to compare between inpatient opioid abusers and non-opioid abusers for Florida hospitals from 2011 to 2017. Florida is one of the eight states labeled as a high-burden opioid abuse state and is an epicenter for opioid use and misuse. A retrospect case-control longitudinal study design on inpatient administrative discharge data across 173 hospitals. Opioid abuse was defined using both ICD-9-CM and ICD-10-CM systems. We found a statistically significant association between opioid abuse diagnosis and total room charge. On average, the opioid abuse status increased the room charges by 8.1%. We also noticed year-to-year variations in opioid abuse had a remarkable influence on hospital finances. We showed that since 2015, the differences are significantly bumped from 4-5% to 13-14% for both room charges and cost, which indicates the financial burden due to opioid abuse becoming more frequent. These findings are important to policymakers and hospital administrators because it provides crucial insight into Florida’s opioid crisis and its economic burden on hospitals.”

  • Line 47 – typo “.,”

Thank you, we have fixed the typo and also have proofread the manuscript.

  • Line 56 – please provide full description of ED abbreviation

Thank you, we moved the full description of the ED abbreviation up to line 56

  • Author contribution – repeated heading and divided into two parts

Thank you, we have updated the Author contribution statement.

Reviewer 2 Report

Dear Authours,

many thanks for submitting this paper and allowing me to review.

The paper is good and of interest but I feel it could be improved.

Line 46 sated the age was from 12 up, I would be keen to see a chronological breakdown as I do not think many under 16 will have opioid use disorder.

As the paper is concentrating on Florida, the Introduction has figures from the US but nothing breaking down to Florida, to demonstrate that the issue is worse there or how it compares to the national data. Again in the Literature review there are figures for expenditure quoted e.g. $18,000 but it is unclear if this is US wide or specific to Florida and how then this compares to one another.

The dataset is very large with over 13 million records. Is there a typo on Line 112 should this be recorded?? again this is a large number of diagnoses (30).

The opioid abuse at <2% is still a reasonable incidence rate from the overall sample.

Line 143 Should capitals be used for the "Greedy Nearest Neighbor Algorithm"?

The figures quoted in results 8.1% and 8.6% with confidence intervals are not clear from the data tables if they are relating to the average over those years this should be a separate entry.

I think the paper should be more reflective on the comparison of data from Florida to the US average as reading the report states Florida is an outlier (one of 8) but fails to reflect this in the text, leaving it unclear if the costs are good per capita or bad and also fails to consider demographics such as age of population which could have an effect on the types of medication required/ prescribed.

I think the paper offers some insight but could be improved and provide more information.

I look forward to seeing a revised version.

Kind regards

Author Response

Reviewer 2

Dear Authors,

many thanks for submitting this paper and allowing me to review.

Thank you so much.

The paper is good and of interest but I feel it could be improved.

Thank you.

Line 46 sated the age was from 12 up, I would be keen to see a chronological breakdown as I do not think many under 16 will have opioid use disorder.

Thank you for your interest. We agree, however, the Substance Abuse and Mental Health Service Association report data aged 12 and up and we added a link to the document in the references. In our study you are right that there are only 202 cases (0.18%) in the opioid abuse group in our final sample. The percentages at each year are between 0.12% to 0.32% from 2011 to 2017. The distribution of the age breakdown is provided in the histogram below.

As the paper is concentrating on Florida, the Introduction has figures from the US but nothing breaking down to Florida, to demonstrate that the issue is worse there or how it compares to the national data. Again in the Literature review there are figures for expenditure quoted e.g. $18,000 but it is unclear if this is US wide or specific to Florida and how then this compares to one another.

Thank you, we have clarified the introduction and identified Florida-related opioids in broad terms. We also inserted more Florida statistics on the economic burden of opioid abuse and overdose deaths.

The dataset is very large with over 13 million records. Is there a typo on Line 112 should this be recorded?? again this is a large number of diagnoses (30).

Thank you, it is not a typo. The state of Florida participates in The Healthcare Cost and Utilization Project program and therefore is asked to record a large number of diagnoses. They record up to 30 and have fields for up to 30. This does not mean that all 30 fields need to be filled. Data directory can be found at https://www.floridahealthfinder.gov/Researchers/OrderData/order-data.aspx

The opioid abuse at <2% is still a reasonable incidence rate from the overall sample.

Thank you.

Line 143 Should capitals be used for the “Greedy Nearest Neighbor Algorithm”?

Thank you. We have capitalized it.

The figures quoted in results 8.1% and 8.6% with confidence intervals are not clear from the data tables if they are relating to the average over those years this should be a separate entry.

Thank you for your feedback. The 8.1% and 8.6% were referring to the overall impact combining the data from all years. To clarify, we have added a row in table 4 to include the overall comparisons.

I think the paper should be more reflective on the comparison of data from Florida to the US average as reading the report states Florida is an outlier (one of 8) but fails to reflect this in the text, leaving it unclear if the costs are good per capita or bad and also fails to consider demographics such as age of population which could have an effect on the types of medication required/ prescribed.

Thank you, we have updated the discussion and introduction to have the introduction focus on Florida while the discussion compares Florida to the US.

I think the paper offers some insight but could be improved and provide more information.

Thank you

I look forward to seeing a revised version.

Thank you

Kind regards

Reviewer 3 Report

General comments 

The article provides a limited finding regarding opioid use in hospitals but not elaborate the possible reasons. Do you believe the problems of abuse opioid use since 2015? It needs to provide strong the evidences to support the hypothesis.   

Specific comments

1.          CONCEPTUAL FRAMEWORK is needed to clearly define the whole picture. Please clearly your study question or hypothesis in the article. 

2.          In the study population, do you have inclusion and exclusion criteria in the case and control group? 

3.          Please clearly define financial items in hospital cost and room charges and percent differences between treatment (opioid abused) and control Groups (non-opioid abused). Why the study classify the surgical and non-surgical group? Patients from the emergency department visits and inpatient stays may contribute the usage the opioid in hospital. Are there the difference of demographics and severity of diseases in the two group? 

4.          How the article clarify measure the abuse opioid use since 2015? It needs to adjust other covariates, which is linked to opioid use. 

5.          In the Discussion, it is needed to explain the reason of increase opioid-related hospital use and stated the scenarios in USA hospitals from the related articles. 

6.          Financial budgets in hospitals is clearly described and do any regulations or guideline from insurance system, it needs to describe the scenarios in detail. 

7.          The article should provide usage of opioid in the distribution of different diseases and surgeries. Specifically, the findings not just mention the increase of opioid in hospitals since 2015 but also find and explain the reasons of opioid usage. From different viewpoints including the types and severity of diseases, patient’s and family member’s require, physician’s prescription, insurance systems etc. was assessed and discussed in the article.               

The title should be clearly defined, because it mentions “Economic Burden” do not equalize the financial cost and room charges. In addition, the article did not accurately economic burden for society or health insurance?  

Author Response

Reviewer 3

General comments 

The article provides a limited finding regarding opioid use in hospitals but not elaborate the possible reasons. Do you believe the problems of abuse opioid use since 2015? It needs to provide strong the evidences to support the hypothesis.   

Thank you, we have added additional explanation and evidence to support our research question and have added.

Specific comments

  1.          CONCEPTUAL FRAMEWORK is needed to clearly define the whole picture. Please clearly your study question or hypothesis in the article. 

Thank you, we have added clarity in the framework and aligned it with our study question.

  1.          In the study population, do you have inclusion and exclusion criteria in the case and control group? 

Thank you, we have added clarity on the inclusion and exclusion and explicitly labeled them. We added “Our inclusion criteria for both control and treatment group were patients admitted to a hospital for treatment between 2011 and 2017. We included in the control group patients without an opioid diagnosis and in the treatment group are patients with at least 1 opioid abuse diagnosis. We excluded any patients with missing data.”

  1.          Please clearly define financial items in hospital cost and room charges and percent differences between treatment (opioid abused) and control Groups (non-opioid abused). Why the study classify the surgical and non-surgical group? Patients from the emergency department visits and inpatient stays may contribute the usage the opioid in hospital. Are there the difference of demographics and severity of diseases in the two group? 

Surgical patients will typically have higher charges than non-surgical patients. U.S.-based studies have identified this as a major point when examining financial costs. Therefore, to address this difference, we included the surgical status (surgical vs. non-surgical) as one of the matching factors when we developed our matching algorithm. In addition, we made a stratified comparison for each surgical status. You made a good point about the emergency department visit that we can look into. The demographical factors (age and gender), inpatient days (length of stay), and the severity of diseases (Elixhauser comorbidity index) were all included in the matching procedures so that the treatment and control groups are similar regarding these factors.

  1.          How the article clarify measure the abuse opioid use since 2015? It needs to adjust other covariates, which is linked to opioid use.

Since 2015, we did not utilize a crosswalk between ICD-9 and ICD-10 but rather examined studies before the Q4 2015 ICD-10-CM change. We have clarified that.  

  1.          In the Discussion, it is needed to explain the reason of increase opioid-related hospital use and stated the scenarios in USA hospitals from the related articles. 

Thank you, we have added the three major reasons for increased opioid-related hospital use in the USA in the Discussion.

  1.          Financial budgets in hospitals is clearly described and do any regulations or guideline from insurance system, it needs to describe the scenarios in detail. 

Thank you. We have added clarity “While the effect of health insurance has increased access to prescribed opioids, providers and insurance companies can reduce the need of opioids through best practices to improve health, closer monitoring of opioid usage, proactive prescribing of mental health services and to drug treatment [34]. Insurance companies and pharmacy benefit managers are in a unique position to monitor that manage opioid usage for the insured since they manage prescriptions claims. Effective policies should be implemented by health plans that identifies appropriate care and prevention strategies and determines the most effective methods for realizing health outcomes to improve population health and optimize reimbursement under policies such as value-based purchasing[35]

  1.          The article should provide usage of opioid in the distribution of different diseases and surgeries. Specifically, the findings not just mention the increase of opioid in hospitals since 2015 but also find and explain the reasons of opioid usage. From different viewpoints including the types and severity of diseases, patient’s and family member’s require, physician’s prescription, insurance systems etc. was assessed and discussed in the article.               

Thank you for your feedback. The primary endpoint of this study is the financial consequences of opioid abuse in Florida hospitals. The findings proved our research question quantitatively that the significance of the issue. I agree with you that it will be valuable to take a close look at the distribution of opioid abuse versus disease groups and find out the reasons that can explain. This is not in the scope of this manuscript but will certainly be in our research agenda for the future.

The title should be clearly defined, because it mentions “Economic Burden” do not equalize the financial cost and room charges. In addition, the article did not accurately economic burden for society or health insurance?  

Thank you, we agree and have updated the title to “The Financial Burden of Opioid-Related Abuse Among Surgical and Non-Surgical Patients in Florida: A Longitudinal Study”

Round 2

Reviewer 1 Report

Review of revised version of the manuscript entitled: “The Economic Burden of Opioid-Related Abuse Among Surgical and Non-Surgical Patients: A Longitudinal Study” authored by Jing Xu, Nazik M.A. Zakari, Hanadi Y. Hamadi, Sinyoung Park, D. Rob Haley, Mei Zhao

In the revised version of the presented manuscript, authors responded to most of my concerns in sufficient way. I regret that authors did not decide to extend the analysis, I encourage authors to perform detailed analysis in the future. In my opinion quality of the manuscript is greatly improved in comparison to first version. However I have some further comments.

Major concerns:

  • Table 4. is a little problematic for me to understand, does it present final results of difference between non-abusers and abusers with respect to treatment and lack of it?
  • It would be nice to see average costs of room charge and hospital for each group in each year if this data is available.

Minor concerns:

  • Line 12-13 – seems like introduction sentence, would be nice to move it to first sentence of the abstract.
  • Line 49 – ED abbreviation introduced two lines above, no need to repeat that
  • Line 136 – typo “.”

Author Response

In the revised version of the presented manuscript, authors responded to most of my concerns in sufficient way. I regret that authors did not decide to extend the analysis, I encourage authors to perform detailed analysis in the future. In my opinion quality of the manuscript is greatly improved in comparison to first version. However I have some further comments.

Major concerns:

  • Table 4. is a little problematic for me to understand, does it present final results of difference between non-abusers and abusers with respect to treatment and lack of it?

Thank you. In our notation, the treatment group is the opioid-abused group, and the control group is the non-abused group. Table 5 presents the comparisons between the two groups for the financial endpoints. The ‘overall’ row compares the two groups across all years for all subjects (the numbers are used in the text), while the other rows compare the two groups by year and surgical status.

  • It would be nice to see average costs of room charge and hospital for each group in each year if this data is available.

Thank you. Please see the information below. The distribution of room charges and hospital cost are provided in Table 3 by year. The distribution shows that for room charges the largest difference between our control and treatment group was in year 2015, while the smallest difference was in year 2014. For hospital costs the largest difference was reported in year 2015 with a 20.58% difference between treatment and control groups, while the smallest difference was in year 2012.

Table 3. Descriptive Statistics of Treatment (Opioid Abuse status) and Control Groups (non-opioid abuse status) Room Charges and Hospital Costs.

Room Charges

Year

Treatment Group (Abuse=1)

Control Group (Abuse=0)

Difference (Treatment – Control)

Percent Change

2011

$12,11.87

$10,513.54

$1,604.33

15.26%

2012

$19,251.52

$17,061.79

$2,189.73

12.83%

2013

$13,967.19

$11,956.54

$2,010.65

16.82%

2014

$7,787.55

$6,912.42

$875.13

12.66%

2015

$16,150.42

$13.732.43

$2,417.98

17.61%

2016

$8,768.44

$7,719.53

$1,048.91

13.59%

2017

$9,628.66

$8,343,53

$1,285.13

15.40%

Hospital Cost

Year

Treatment Group (Abuse=1)

Control Group (Abuse=0)

Difference (Treatment – Control)

Percent Change

2011

$2,484.52

$2,153.24

$331.29

15.39%

2012

$3,568.87

$3,178.65

$390.22

12.28%

2013

$2,462.00

$2,134.73

$327.27

15.33%

2014

$1,318.17

$1,173.28

$144.89

12.35%

2015

$2,690.27

$2,231.16

$459.11

20.58%

2016

$1,505.16

$1,320.95

$184.21

13.95%

2017

$1,456.34

$1,253.02

$203.32

16.23%

Minor concerns:

  • Line 12-13 – seems like introduction sentence, would be nice to move it to first sentence of the abstract.

Thank you. We agree and have moved it

  • Line 49 – ED abbreviation introduced two lines above, no need to repeat that

Thank you. We have removed the duplication. 

  • Line 136 – typo “.”

Thank you. We have removed the period.

Reviewer 2 Report

Many thanks

you have addressed the issued I had made in my earlier review.

Best wishes

Author Response

 Thank you.

Reviewer 3 Report

Please use the multivariate analysis for financial items in hospital cost and room charges after adjusted for covariates? 

Please highlight to discuss the policy of medical insurance to reduce Opioid-Related abuse in the USA. 

Please to clearly define the framework developed from y Leslie, Ba. 

The format of references should be fitted the requirement of the Journal. 

Author Response

Please use the multivariate analysis for financial items in hospital cost and room charges after adjusted for covariates? 

Thank you for your recommendation. I agree with you that multivariate analysis may be another option to analyze the data. In our thinking process of data analysis, our primary interest is the room charges since that’s the financial burden to the payers. Hospital cost was the secondary interest as a measure of hospital performance. Therefore, we did the analysis separately, which is also appropriate statistically. The results may be more significant after the correlations were accounted for, but I don’t expect the conclusions to change. 

Please highlight to discuss the policy of medical insurance to reduce Opioid-Related abuse in the USA. 

Thank you, we have added “The effect of health insurance on opioid abuse and misuse has been under debate.  Some believe that access to health insurance increased access to prescribed opioids.  However, policies are increasingly being implemented for insurance companies to more closely monitor opioid usage and to educate providers on proper opioid prescription policies [40]. For example, under the ACA the expansion of health insurance was associated with the meaningful reductions in opioid related hospital use health plans implemented improved and proactive utilization management care for opioid use disorder among the Medicaid expansion segment [41].”

Please to clearly define the framework developed from y Leslie, Ba. 

Thank you, we have added additional clarity of the framework. “The framework focuses on the path of patients with pain conditions. These patients may begin to used prescribed opioid medication and may become addicted which can result in an opioid-abuse diagnosis. Treatment is typically sought out and initiated in the ED and leads to further healthcare services including but not limited to inpatient services (eg, hospitalizations and residential rehabilitation services).”

The format of references should be fitted the requirement of the Journal. 

Thank you we have updated the references to the MDPI ACS Journals format style indicated in the journal author guideline in endnote.
